# Genome-Wide Identification, Characterization, and Expression of *TCP* Genes Family in Orchardgrass

**DOI:** 10.3390/genes14040925

**Published:** 2023-04-16

**Authors:** Cheng Wang, Guangyan Feng, Xiaoheng Xu, Linkai Huang, Gang Nie, Dandan Li, Xinquan Zhang

**Affiliations:** College of Grassland Science and Technology, Sichuan Agricultural University, Chengdu 611130, China; wangcheng2709@163.com (C.W.); feng0201@sicau.edu.cn (G.F.); xuxiaoheng429@163.com (X.X.); huanglinkai@sicau.edu.cn (L.H.); nieg17@sicau.edu.cn (G.N.); lidandan@sicau.edu.cn (D.L.)

**Keywords:** *Dactylis glomerata*, *TCP* gene family, floral development, tillering, expression analysis

## Abstract

Plant-specific TCP transcription factors regulate several plant growth and development processes. Nevertheless, little information is available about the TCP family in orchardgrass (*Dactylis glomerata* L.). This study identified 22 *DgTCP* transcription factors in orchardgrass and determined their structure, phylogeny, and expression in different tissues and developmental stages. The phylogenetic tree classified the *DgTCP* gene family into two main subfamilies, including class I and II supported by the exon–intron structure and conserved motifs. The *DgTCP* promoter regions contained various cis-elements associated with hormones, growth and development, and stress responses, including MBS (drought inducibility), circadian (circadian rhythms), and TCA-element (salicylic acid responsiveness). Moreover, *DgTCP9* possibly regulates tillering and flowering time. Additionally, several stress treatments upregulated *DgTCP1, DgTCP2, DgTCP6, DgTCP12,* and *DgTCP17*, indicting their potential effects regarding regulating responses to the respective stress. This research offers a valuable basis for further studies of the *TCP* gene family in other Gramineae and reveals new ideas for increasing gene utilization.

## 1. Introduction

The plant-specific TEOSINTE BRANCHED 1/CYCLOIDEA/PROLIFERATING CELL FACTOR (TCP) gene family was first discovered in 1999 [1]. The acronym TCP comes from four genes in three species: T (*TB1* from maize [*Zea mays*]) [2], *C* (*CYC* from snapdragon [*Antirrhinum majus*]) [3], and P (*PCFs* from rice [*Oryza sativa*]) [4]. Members of the TCP family have 59 amino acids and an atypical basic helix–loop–helix (bHLH) motif at the N-terminus called the TCP domain which is responsible for protein–protein interactions, nuclear protein localization, and DNA binding [1,4]. Moreover, the TCP domain classifies these proteins into two groups: class I and II [5]. The clearest distinction between class I and II is the basic region of the TCP domain where class I members lost four amino acids. Class II is divided into CYC/TB1 and CIN subclades [5,6]. Arginine-rich motifs with 18–20 residues (R domain) are usually found in class II [1].

Many TCP proteins are critical in several plant biological processes [5,7], including hormone biosynthesis, signaling transduction, flower development, leaf development and senescence, lateral branching, circadian rhythm, seed germination, defence response, and cell proliferation and differentiation [7,8,9,10,11,12,13,14,15,16,17,18]. For instance, AtTCP14 and AtTCP15 class I proteins regulate leaf morphology and embryonic development during seed germination through the gibberellin (GA) signaling pathway [19,20]. *AtTCP20* acts upstream of *AtTCP9* to regulate leaf senescence via the jasmonic acid (JA) signal pathway [21]. *AtTCP16* is necessary for pollen development in developing microspores [22]. In *Arabidopsis thaliana*, five miR319a targets in CIN class II gene subgroups (*AtTCP2*, *AtTCP3*, *AtTCP4*, *AtTCP10,* and *AtTCP24*) were confirmed important in leaf growth and morphogenesis [9,12]. *AtTCP4* is essential for the natural life activity of petals, the multifaceted regulation of JA and auxin (IAA) synthesis, the senescence of ripe leaves, and the age-dependent decomposition of leaf photosynthetic complexes [21,23,24,25,26]. In the CYC/TB1 subclade of class II, *LjCYC1* and *LjCYC3* (in *Lotus japonicus*) and *AmCYC* (in snapdragon) produce a marked effect in floral development [3,27]. Moreover, the *A thaliana AtTCP1* and *GhCYC2* from *Gerbera hybrida* control symmetrical petal growth in flowers [28,29]. *ZmTB1* in maize, *AtBRC1* and *AtBRC2* in *A thaliana*, and *OsTB1* in rice modulate branching by negatively regulating axillary bud growth [11,30,31]. Besides, *TCPs* regulate the circadian clock and plant morphogenetics. *TCP2*, *3*, *11*, and *15* combine with the TGGGC (C/T) element to interact with various compositions of the core circadian rhythm, thus mediating the *Arabidopsis* circadian clock. Additionally, *TCP20* and *22* improve the circadian clock [10,32].

Environmental stresses can affect plant growth and development [33], and some TCP family members have been shown to respond to environmental changes. For instance, overexpressing *OsTCP19* induces the typical genes from abscisic acid (ABA), methyl jasmonate (Me-JA), ethylene (ET), IAA, cytokinins (CK), and other signaling pathways in rice. These genes reduce reactive oxygen species, the accumulation of fat droplets, and water loss in transgenic plants, thus improving their tolerance to high salt and mannitol treatments [34]. In rice, downregulating *OsTCP21* and *OsPCF6* enhances tolerance to cold stress by altering the scavenging of reactive oxygen species. Similarly, *OsPCF5* and *OsPCF8* improve tolerance to cold stress [35,36]. Binding *OsPCF2* to the OsNHX1 promoter improves salt and drought tolerance [37]. In *A thaliana, TCP20* associates with NLP6/7 to modulate signal transduction and nitrate assimilation [38]. Several *TCP* genes have also been reported in other species. For example, four miR319 target genes, including *AsPCF5*, *AsPCF6*, *AsPCF8,* and *AsTCP14,* were downregulated in drought and salt tolerance creeping bentgrass (*Agrostis stolonifera*) [39,40]. Some *TCP* genes in common bean (*Phaseolus vulgaris*) and *PeTCP10* in moso bamboo (*Phyllostachys edulis*) enhance tolerance to salt stress [41,42]. Besides, several *GhTCP* genes in cotton (*Gossypium hirsutum*) were upregulated under drought salt and heat stress [17]. Although the *TCP* gene family plays a major role in regulating plant life processes, there are no reports of their functioning in orchardgrass (*D glomerata*).

Orchardgrass is a widely cultivated perennial forage grass that is native to central and western Europe, the temperate regions of Asia, and North Africa [43]. *D glomerata* has a high nutrient content and is among the four major global economic perennial grasses. It establishes fast, recovers quickly after mowing, and has high shade, drought, and barren tolerance [44]. Many valuable genes control orchardgrass development and abiotic stress response. Therefore, this study identified *TCPs* in orchardgrass through synthetic analysis of the *D. glomerata* genome (gene structure, conserved motif composition, chromosomal location, and phylogenetic characteristics). Preliminary predictions of *DgTCP* gene evolution involved the analysis of phylogenetic and gene duplication events and collinearity with other plants. Additionally, *TCP* expression was assessed in different tissues, developmental stages, and abiotic stresses. The results of this study will be helpful in elucidating orchardgrass adaptation to different environments and may reveal the functions of *DgTCPs*.

## 2. Material and Methods

### 2.1. Identification of Dactylis glomerata TCP Genes

Firstly, the TCP domain (PF03634) HMM profile, which was obtained from the Pfam database (http://pfam.xfam.org/, accessed on 1 March 2022) [45], was the reference for identifying TCP genes from the Dactylis glomerata genome (http://orchardgrassgenome.sicau.edu.cn/, accessed on 20 January 2022) using HMMER 3.0 software (E-value cutoff of 0.01) [46]. Secondly, PFAM was used to further analyze all candidate genes. The confirmed *TCPs* were aligned with Clustal X2.0 [47], and redundant sequences were discarded. Finally, the physicochemical properties of the DgTCP protein, including protein length, CDS length, isoelectric point, and molecular weight, were determined using ProtParam (http://web.expasy.org/protparam/, accessed on 3 March 2022) [48].

### 2.2. Phylogenetic Analysis and Classification of DgTCP Genes 

The sequences of *A thaliana TCP* genes were obtained from TAIR (https://www.arabidopsis.org/, accessed on 12 February 2022) [49]. Next, 22 rice and 21 *Brachypodium distachyon* TCP protein sequences were obtained from Plant TFDB (http://planttfdb.cbi.pku.edu.cn/, accessed on 6 March 2022) [50]. Multiple alignments of the selected TCP sequences were performed using Clustal X2.0 [47]. Based on the neighbor-joining (NJ) method and 1000.0 replicates for bootstrap node support, the phylogenetic tree of orchardgrass, rice, *B distachyonwere*, and *A thaliana* were constructed using MEGA7.0 [51] and then beautified via the iTOL website (https://itol.embl.de/itol.cgi/, accessed on 14 April 2022).

### 2.3. Gene Structure and Motif Analysis

The exon–intron structures of *DgTCPs* were generated based on available genomic information and coding sequences from the Gene Structure Display Server (GSDS 2.0, http://gsds.cbi.pku.edu.cn/, accessed on 6 May 2022) [52]. The online Multiple Expectation Maximization for Motif Elicitation (MEME) software (http://meme-suite.org/, accessed on 11 May 2022) was used to identify the conserved DgTCP proteins motifs (considering ten maximum motifs and default settings) [53].

### 2.4. Putative Promoter Cis-Acting Element Analysis

The nucleotide sequences of *DgTCPs* were acquired from the orchardgrass genome database (http://orchardgrassgenome.sicau.edu.cn/, accessed on 20 January 2022). The 2000 bp region upstream of all *DgTCPs* was considered the promoter sequence, and the cis-acting promoter elements were appraised via PlantCARE (http://bioinformatics.psb.ugent.be/webtools/plantcare/html/, accessed on 19 June 2022) [54]. The putative cis-acting elements are classified into plant hormone responses, growth and development, and biotic and abiotic stress responses.

### 2.5. Chromosomal Mapping and Synteny Analysis

The chromosomal position information of each *TCP* gene was retrieved from orchardgrass genome annotations. MapGene2Chrome (MG2C, http://mg2c.iask.in/mg2c_v2.0/, accessed on 26 June 2022) was used to describe the location of the *TCPs* on the chromosomes. Next, *DgTCP* gene duplication was analyzed using MCScanX and default parameters [55]. Using the Dual Synteny Plotter of TBtools, we mapped the *TCP* gene collinearity between *D glomerata*, *A thaliana*, *O sativa, Sorghum bicolor, Hordeum vulgare,* and *B distachyon* (https://github.com/CJ-Chen/TBtools, accessed on 21 April 2022) [56]. The genome data of *O sativa, S bicolor, B distachyon,* and *H vulgare* were downloaded from the JGI Genome Portal (https://genome.jgi.doe.gov/portal/, accessed on 12 March 2022), and *A thaliana* was downloaded from TAIR (https://www.arabidopsis.org/, accessed on 12 February 2022).

### 2.6. Plant Material and Treatments

Seeds of “Baoxing” orchardgrass were grown in a growth chamber at a 22 °C/14 h (day) and 20 °C/10 h (night) cycle. One week after germination, the seedings were irrigated with 1/2 Hogland solution. When the seedlings reached the third to fourth leaf, they were treated with 1/2 Hoagland solution containing 20% polyethylene glycol (PEG6000), 250 mM sodium chloride (NaCl), 100 μM methyl jasmonate (Me-JA), 200 mM sodium bicarbonate (NaHCO_3_), 100μM abscisic acid (ABA), and 100 μM salicylic acid (SA). The leaves for each treatment were collected separately at 0, 1, 3, 6, 12, and 24 h after treatment, immediately frozen in liquid nitrogen, and then stored at −80 °C for qRT-PCR.

### 2.7. Expression Profiles of DgTCP Family Members

The expression patterns of *TCPs* in the root, stem, leaf, spike, and flower were obtained from the orchardgrass genome database (Appendix A) [46]. Furthermore, the expression patterns of the floral bud developmental stages before vernalization (BV), vernalization (V), after vernalization (AV), before heading (BH), and heading (H) of the late-flowering variety “Baoxing” and early-flowering variety “Donata” were obtained from the RNA-seq data (Appendix A) [57]. The RNA-seq data of the *TCP* genes in four tissues from varieties D20170203 (low-tillering) and AKZ-NRGR66 7 (high-tillering) were obtained from Xu et al. (Appendix A) [58]. The heat maps of the expression patterns were produced using TBtools [56]. 

### 2.8. Expression of 14 Selected DgTCP Genes in qRT-PCR

The total RNA of the samples under different treatments was extracted using the Hipure HP plant RNA mini kit (Magen, Guangdong, China). First-strand cDNA was synthesized using the MonScript^TM^ RTIII ALL-in-One Mix with dsDNase kit (Monad, Suzhou, China) following the manufacturer’s instructions. Primers for the 14 *DgTCPs* were designed using Primer 5.0 software (Appendix A), and qRT-PCR was performed using the MonAmpTM SYBR^®^ green qPCR Mix (Monad) on the Bio-Rad CFX96 instrument. GAPDH was the internal reference gene for normalization [43], and the relative gene expression levels were evaluated by the 2^−ΔΔCt^ method [59]. All qRT-PCR assays were performed with three biological and technical replicates. 

## 3. Results

### 3.1. Identifying TCP Genes in Orchardgrass

Twenty-two genes were retrieved from the orchardgrass genome and designated as *DgTCP1*–*DgTCP22* based on their chromosomal positioning. The protein sequence length, CDS length, molecular weight, isoelectric point (pI), and gene location of the 22 *DgTCPs* are captured in Table 1 and Appendix A. The smallest protein was 17,433.51(*DgTCP1*), and the biggest was 47,382.56 kDa (*DgTCP16*). The pI ranged from 5.09 (*DgTCP7*) to 9.92 (*DgTCP11*), and the protein lengths were 165 (*DgTCP17*) to 454 (*DgTCP6*) aa.

### 3.2. Phylogeny and Classification of the DgTCP Proteins

Based on the phylogenetic tree of 22 orchardgrass, 22 rice, 21 *B distachyon*, and 24 *A thaliana*, TCP proteins were constructed using the neighbor-joining (NJ) method to clarify the phylogenetic relationships and evolutionary history of the *TCP* gene family (Figure 1). Two classical subfamilies, class I and class II, were identified from the topology of the NJ tree and *A thaliana* classification. Eleven *DgTCPs* belong to class I (PCF or TCP-P), and the other 11 belong to class II (TCP-C) (Figure 1). The class II group is further divided into CYC/TB1 (4 *DgTCPs*) and CIN subclasses (7 *DgTCPs*) (Figure 1). Thus, we performed multiple sequence alignments on the TCP domains of all *DgTCP* members to comprehend the phylogenetic relationships of the *DgTCPs*. The TCP domain comparison and phylogenetic analysis indicated that orchardgrass TCP proteins have class I (PCF) and class II (CIN and CYC/TB1) groups (Figure 1 and Figure 2). Class I proteins lack four amino acids at their basic domain compared with class II proteins.

### 3.3. The DgTCP Gene Structure and Protein Motif 

The structural characteristics of all DgTCPs were analyzed to comprehend the evolution of the *TCP* gene family in orchardgrasss (Figure 3b). All class I *DgTCP* genes, except *DgTCP5* and *DgTCP22*, lack introns. In class II, all CIN genes possess one or two introns, while CYC/TB1 genes lack introns.

Figure 3c shows 10 conserved motifs of the 22 DgTCP proteins which were identified using MEME to reveal the structural characteristics of orchardgrass *TCP*. The amino acid sequence for each motif (Appendix A) shows that the conserved motifs have 6–42 amino acids. All DgTCP protein contain motifs 1 and 2. Moreover, DgTCP proteins from the same subfamilies contain similar motifs. For instance, members of clade PCF contain motif 3, while the clades CIN and CYC/TB1 lack motif 3. All clade CIN members contain motifs 7 and 9, while PCF members contain motifs 5, 6, and 8.

Additionally, some motifs, such as motifs 4 and 10, are shared by two classes. These results indicate that the motifs present only in some subgroups may be associated with unique functions. Nevertheless, the unique functions of these motifs in the plant life cycle have not been identified and need to be explored further.

### 3.4. Chromosomal Localization, Gene Duplication, and Synteny Analysis

The 22 orchardgrass *TCP* genes are randomly distributed on 7 chromosomes (Figure 4). Chromosome 4 contained six *TCP* genes, and chromosomes 3 and 5 had five *TCP* genes each. Three *TCP* genes mapped to chromosome 1, but only one mapped to chromosomes 2, 6, and 7.

The duplication event is important for analyzing the evolution and expansion of the gene families. The orchardgrass genome has five pairs of segmental duplicates (Appendix A), including *DgTCP2/DgTCP14*, *DgTCP3/DgTCP5*, *DgTCP4/DgTCP9, DgTCP6/DgTCP12*, and *DgTCP7/DgTCP22* (Figure 5, linked with red lines).

The five comparative syntenic maps show the evolutionary relationships among *TCPs* in different species, including *A thaliana* (dicotyledonous plant)*, O sativa, S bicolor, B distachyon,* and *H vulgare* (Figure 6). The homologous pairs between *D glomerata* and the 5 species were 33 (*O*), 31 (*S bicolor*), 30 (*B distachyon*), 23 (*H vulgare*), and 5 (*A thaliana*) (Appendix A). These results indicate that the *TCPs* in the monocotyledons are highly conserved and homologous.

### 3.5. Putative Cis-Acting Elements of Orchardgrass DgTCPs

The cis-elements in promoters are essential for transcriptional regulation and gene function analysis. Therefore, to provide further insight into the gene functions and regulation mechanisms of *DgTCP* genes, 93 cis-elements possibly involved in phytohormone response, plant growth and development, and stress response were identified to unravel the functions and regulatory mechanisms of *DgTCP* genes (Appendix A). The TATA- and CAAT-box had the most cis-elements among the 22 *DgTCPs* (Appendix A). Interestingly, the AACA-motif, MBSI, HD-Zip 1, circadian, and AuxRR-core only existed in 1 of 22 *DgTCPs* (Figure 7), indicating their likely unique roles in those genes and, by extension, the regulatory pathways and processes involving those genes.

The promoter regions of one and three *DgTCPs* are two cis-elements (AACA-motif and GCN4-motif) that participate in endosperm expression. Besides, NON-box and CAT-box were associated with meristem expression plant growth and development. The seed-specific regulatory element (RY element) and zein metabolism regulatory element (O2 site) were identified in six and nine *DgTCPs,* respectively. In addition, a circadian control element (circadian), a flavonoid biosynthetic regulation element (MBSI), a cell cycle regulation element (MSA-like), and a palisade mesophyll cells regulatory element (HD-Zip 1) were also discovered in the promoter regions of *DgTCPs* (Figure 7). In several hormone-related cis-elements, the salicylic acid (TCA element), the auxin-responsive element (AuxRR core and TGA element), the gibberellin-responsive element (GARE motif, TATC-box, and P-box), the Me-JA-responsive element (CGTCA motif and TGACG motif), and the ABA-responsive element (ABRE) were found in the promoter region of 9, 9, 15, 19, and 19 *DgTCP* genes, respectively (Figure 7). In addition, the *DgTCP* promoters contained several cis-elements that were related to several stresses (drought, anaerobic induction, and low temperature) (Figure 7).

### 3.6. Expression Profiles of DgTCPs in Different Tissues and Developmental Stages

The expression profiles of 2, 5, and 14 *DgTCP* genes were the highest in the leaf, spike, and stem, respectively (Figure 8a). Moreover, *DgTCP1* and *DgTCP9* had higher transcription levels in the flowers, revealing that these genes might be important for the growth of different orchardgrass tissues.

The expression patterns of the early- (Baoxing) and late-flowering (Donata) cultivars were analyzed at five flower bud development stages to identify their potential physiological functions in flowering. In most developmental stages, the expression of *DgTCP9* and *DgTCP6* was higher in “Baoxing” than “Donata” (Figure 8b). *TCP15* expression was similar in “Baoxing” and “Donata” before, during (downregulated), and after vernalization (upregulated). *TCP18* was significantly upregulated during vernalization in “Baoxing” but showed no change in “Donata”. However, it was upregulated in “Baoxing” and “Donata” during the late vernalization stage and similar in “Baoxing” and “Donata”. From after vernalization to the heading stage, *DgTCP2*, *DgTCP4*, *DgTCP16*, *DgTCP17*, and *DgTCP21* were significantly upregulated during the before heading stage in “Baoxing” and the heading stage in “Donata”.

Gene expression data were retrieved from four tissues of low- (D20170203) and high-tillering (AKZ-NRGR667) orchardgrass to determine the roles of *DgTCPs* in regulating growth, and development. All *DgTCPs* were differentially expressed in the four tissues, contrary to their expression under normal conditions (Figure 9). In tiller buds, over half of the *DgTCPs* were highly expressed in the low- (D20170203) and high-tillering (AKZ-NRGR667) varieties. The expression of *TCP1* was significantly higher in D20170203 than AKZ-NRGR667 in the four tissues. However, the expression of *TCP20* was higher in the leaves of AKZ-NRGR667 than in D20170203 but similar in other tissues. The expression of *TCP10* was higher in the tiller bud of D20170203 than in AKZ-NRGR667 but similar in other tissues. Interestingly, the expression of *TCP9* was higher in the tiller buds of D20170203 than those of AKZ-NRGR667 but lower in the leaves of D20170203. Thus, *DgTCP1* and *DgTCP10* probably inhibit tiller bud development in the two varieties in a differential pattern, ultimately resulting in different phenotypes.

### 3.7. The Expression of DgTCP Genes under Six Abiotic Stresses

Figure 10 shows the expressions of 14 *DgCTPs* in “Baoxing” under six abiotic stresses (drought, salt, alkali, Me-JA, ABA, and SA). Salt stress suppressed the expression of two genes (*DgTCP8* and *DgTCP18*) throughout the salt time points and downregulated eleven genes in the early stages (1–3 h) of salt stress. Drought stress upregulated 13 *DgTCPs,* and the highest values ranged from 1.13-fold (*DgTCP3*) to 16.89-fold (*DgTCP1*). Additionally, six genes (*DgTCP2*, *DgTCP12*, *DgTCP15*, *DgTCP16*, *DgTCP17,* and *DgTCP18*) showed significantly higher expression 1 h after treatment with an alkali solution. In contrast, other *DgTCPs* were upregulated at 6 h of alkali treatment, while *DgTCP8* was suppressed at all time points. Nine *DgTCPs* were highly induced under ABA treatment, and *DgTCP12* displayed the highest expression. Nevertheless, ABA treatment inhibited *DgTCP3*, *DgTCP8*, *DgTCP9*, *DgTCP16,* and *DgTCP19* at all time points. Me-JA treatment upregulated five (*DgTCP1*, *DgTCP2*, *DgTCP6*, *DgTCP10,* and *DgTCP17*), one (*DgTCP8*), and two genes (*DgTCP3*, *DgTCP12*), which peaked at 3, 6, and 12 h, respectively. These genes were upregulated ranging from 1.26-fold (*DgTCP18*) to 20.16-fold (*DgTCP12*). In contrast, Me-JA showed no observable regulation in four genes (*DgTCP4*, *DgTCP8*, *DgTCP15*, and *DgTCP16*) but downregulated *DgTCP9* and *DgTCP19.* Furthermore, SA treatment suppressed the expression of *DgTCP3*, *DgTCP9,* and *DgTCP19* across the time points, with six (*DgTCP1*, *DgTCP2 DgTCP12*, *DgTCP15*, *DgTCP16,* and *DgTCP17*) and three genes (*DgTCP4*, *DgTCP6*, and *DgTCP18*) peaking at 6 and 24 h, respectively. Six stress treatments upregulated *DgTCP1, DgTCP2, DgTCP6, DgTCP12,* and *DgTCP17,* but *DgTCP* levels varied under different stresses and time points when combined with the stress expression pattern data.

## 4. Discussion

The plant-specific TEOSINTE BRANCHED 1/CYCLOIDEA/PROLIFERATING CELL FACTOR (*TCP*) gene family are crucial plant-specific transcription factors with various functions in many processes, including hormone biosynthesis, flower development, leaf development, lateral branching, and defence response. To date, the *TCP* gene family has been identified in many plants, such as switchgrass (*Panicum virgatum*) [60], maize [61], *Arabidopsis,* and rice [62]. However, a comprehensive report of the *TCP* gene family in high-quality forages, such orchardgrass, is lacking.

This research identified 22 *TCP* genes in the orchardgrass genome [46]. All the corresponding *DgTCP* proteins have a highly conserved TCP domain (motifs 1 and 2). Thus, *DgTCPs* possibly have similar DNA-binding and protein–protein interaction patterns [1,4]. Moreover, sequence alignment and phylogenetic analysis revealed that the 22 *DgTCPs* are divided into two major subclasses (Figure 2 and Figure 3), which is consistent with previous results [5]. Each subclass contained *TCP* genes from *B distachyon*, rice, and *Arabidopsis.* Furthermore, the orchardgrass *TCP* genes are closely related to the *TCPs* of rice and *B distachyon*, indicating that they evolved from a common ancestor as Gramineae. These results show that many *TCP* genes from the same ancestor possibly experienced different differentiation patterns at different lineages. Moreover, *DgTCPs* in the same class and subclass had similar exon–intron structures (Figure 3b) and relatively conserved motifs (Figure 3c), further supporting the close evolutionary relationships between *DgTCPs.*

The number of *DgTCPs* was higher than those in moso bamboo (16) [63], grapevine (*Vitis vinifera*) (17) [64], strawberry (*Fragaria vesca*) (19) [65], and *Sorghum* (20) [66], In contrast, the number of *DgTCPs* was lower than the number in *A thaliana* (24) [62], maize (46) [61], soybean (*Glycine max*) (54) [67], and tobacco (*Nictiana tabacum*) (61) [68]. Tandem, segmental, and whole-genome duplication are important sources of the functional diversity and evolution of gene families [69]. Previous research showed that *D glomerata* experienced whole-genome replication events [46]. This study identified five segmental repeat gene pairs in the *DgTCP* gene family and no tandem duplication (Figure 5). Segmental duplication was more beneficial for expanding and evolving the *D. glomerata TCP* gene family. These results are similar to those described in *A thaliana* and rice, indicating that *TCP* duplication in plant genomes possibly has a common mechanism [62,70].

Abiotic stresses affect plant growth and development, quality, and yield [33], and *TCP* genes are broadly involved in the regulatory processes of plant life [71]. Therefore, exploring the potential functions of *TCP* genes in orchardgrass under different abiotic stresses is necessary. In this study, salt and drought treatments upregulated more than half of the identified *DgTCPs*, similar to the results from rapeseed (*Brassica napus*) [72], cotton [17], and switchgrass [60]. Nonetheless, ABA treatment inhibited five *DgTCPs* at all time points. The ABA signal transduction pathway is significant for stress response [73].

Moreover, *TCPs* interact with other genes in JA biosynthesis to influence growth, development, and abiotic stress responses. For instance, *TCP4* encodes the enzyme that catalyzes a crucial step in JA synthesis by positively regulating the *LOX2* gene [9]. Deactivating *TCP4* in plants downregulates *LOX2*, thus reducing JA synthesis and increasing plant sensitivity to stress [9]. The expression patterns of orchardgrass *TCP* genes were diverse after Me-JA treatment (Figure 10). For example, Me-JA treatment lowered the expression of *DgTCP16* of the *AtTCP4*-like gene, which may be because the promoter region of *TCP16* lacks Me-JA-related cis-elements (Figure 7).

In *A thaliana*, *TCP8* and *TCP9* combine to the TCP-promoter binding site of the SA biosynthesis gene *ICS1*, thus enhancing *ICS1* expression [74]. Additionally, SA treatment increased the expression of many cis-elements related to SA in the promoter regions of the *DgTCP* genes (Figure 7), including *DgTCP2*, *DgTCP4*, *DgTCP12*, and *DgTCP18* (Figure 10). Therefore, *DgTCP* genes may play a role in SA transduction. These results suggest that Dg*TCP* genes are essential for plants to cope with abiotic stress.

Gene function can be inferred from the expression profile of that gene [75]. Thus, this research inferred the functions of 22 *DgTCPs* using their expression patterns in five tissues (Figure 8a). The results showed different expression profiles of the 22 *DgTCPs* in five tissues, indicating that orchardgrass *TCPs* might be related to the development of different tissues. The highest expression of several *DgTCPs*, such as *DgTCP3*, *DgTCP4*, *DgTCP15*, and *DgTCP19*, occurred in the stem. Several *TCP* genes are highly expressed in the stem, including 60 in cotton [17], 11 in rapeseed [72], and 9 in soybean [67]. Some *DgTCP* genes, such as *DgTCP*1 and *DgTCP9*, are highly expressed in flowers, indicating that *DgTCP* genes possibly participate in flowering. As with this study, 16 and 12 highly expressed *TCP* genes were reported in rapeseed and soybean [67,72]. Moreover, most duplicate gene pairs had the same functions and similar expression patterns, except *DgTCP4*, *DgTCP9*, *DgTCP7*, and *DgTCP22*, which showed different expression profiles. This diverse expression may be related to differences in the evolution of duplicate genes or upstream regulatory mechanisms, causing functionalization in one of the duplicate genes.

Next, we compared the potential role of *TCP* genes in regulating flowering time at the five stages of late- and early-flowering orchardgrass (Figure 8b). Flowering is crucial for the growth and development of Gramineae, and variations in flowering time directly affect orchardgrass quality and yield. Moreover, vernalization is a critical way to control flowering time and floral organ development [57]. Thus, the unique expression profiles of *DgTCP15* and *DgTCP18* at different floral bud developmental stages suggests that these genes may regulate flowering time through the vernalization pathway. For example, *A thaliana* plants that overexpress *AtTCP23* have the late-flowering phenotype [76]. In this study, *DgTCP6* was expressed at higher levels in “Donata” than in “Baoxing” from the before vernalization stage to the heading stage of late-flowering “Donata” and early-flowering “Baoxing”. *DgTCP6* and *AtTCP23* belong to the same branch on the evolutionary tree, indicating that they are homologous genes. This alignment implies that *DgTCP6* and *AtTCP23* have similar functions. Thus, a high expression of *DgTCP6* promotes the late-flowering phenotype in “Donata”. Furthermore, the *DgTCP9* gene, which has a similar expression pattern as *DgTCP6* (Figure 8b), may also have a similar flowering regulatory function. Moreover, *AtTCP4* and *AtTCP13*, *AtTCP7* induce early flowering by directly acting on the AP1 promoter to improve its transcript activation ability and activating the transcription expression of the flowering integration gene *SOC1*, respectively [77,78]. The three *TCPs* were grouped with *DgTCP16* and *DgTCP21, DgTCP17,* respectively (Figure 1). Additionally, from the before heading stage to the heading stage, the earlier upregulation of *DgTCP16*, *DgTCP17,* and *DgTCP21* in “Baoxing” relative to “Donata” indicates that they influence early flowering in “Baoxing” and reflect the functions of *AtTCP4* and *AtTCP13, AtTCP7* in *A thaliana*. Altogether, the diverse expression of *DgTCPs* at the five floral bud stages in the different cultivars indicates their regulatory role in orchardgrass flowering.

Finally, the roles of *DgTCP1* in tillering were analyzed in the respective cultivars. Tillering is an important agronomic trait in forage crops as it determines the seed yield and aboveground biomass of forage grasses [79,80]. In this study (Figure 9), the tissue-specific expression patterns of *DgTCP1* in the two forage varieties revealed that a high expression of *DgTCP1* may suppress tillering. Moreover, *OsTCP19*, a *DgTCP9* homologous gene (Figure 1), negatively regulates rice tillering by inhibiting *DLT*, which promotes tillering [81]. The unique expression of *DgTCP9* in high- and low-tillering materials indicates that *DgTCP9* possibly confers low-tillering in “D20170203”. In summary, *DgTCPs* might be important for tiller development; thus, they require further experimental verification.

## 5. Conclusions

This study identified 22 *DgTCPs* from the whole genome of *D glomerata*. Phylogenetic characteristics divided the 22 *DgTCPs* into class I and II subfamilies. The study also revealed the protein sequence length, CDS length, pI, and molecular weight of the proteins predicted from the 22 *DgTCP* genes. Furthermore, we identified many cis-elements in the *DgTCP*-promoter sequences, revealing a complex regulatory network that possibly controls *DgTCP* genes. The 22 *DgTCP* genes contained five pairs of segmental repeat genes distributed on seven chromosomes, indicating that segmental duplication was the primary mechanism for *DgTCP* gene expansion.

Furthermore, the expression of the *DgTCPs* under various abiotic stresses at different stages (tiller bud and floral bud) and tissues suggested that many *DgTCP* genes regulate stress tolerance and development in orchardgrass. Specifically, *TCP9* probably regulates flowering time, tiller number, and drought stress in *D glomerata*. This genome-wide analysis of orchardgrass is significant for identifying new *DgTCP* genes with novel functions and provides a foundation for breeding high-quality orchardgrass varieties and the functional validation of *DgTCP* genes in the future.

## Figures and Tables

**Figure 1 genes-14-00925-f001:**
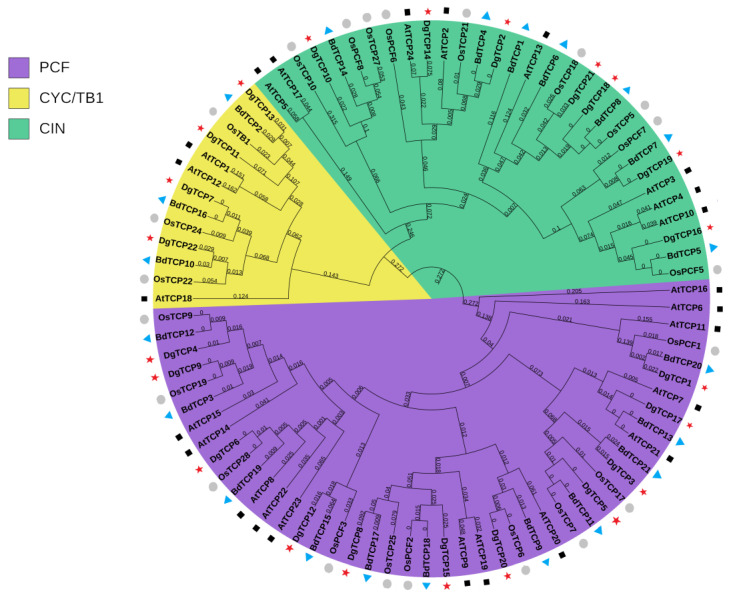
An unrooted phylogenetic tree containing TCP proteins from orchardgrass, rice, *A thaliana*, and *B distachyon*. Green shading, CIN subclass; yellow shading, CYC/TB1 subclass; purple shading, PCF subclass. The grey circle, red pentagram, black square, and blue triangle represent the rice, orchardgrass, *A thaliana*, and *B distachyon TCPs,* respectively.

**Figure 2 genes-14-00925-f002:**
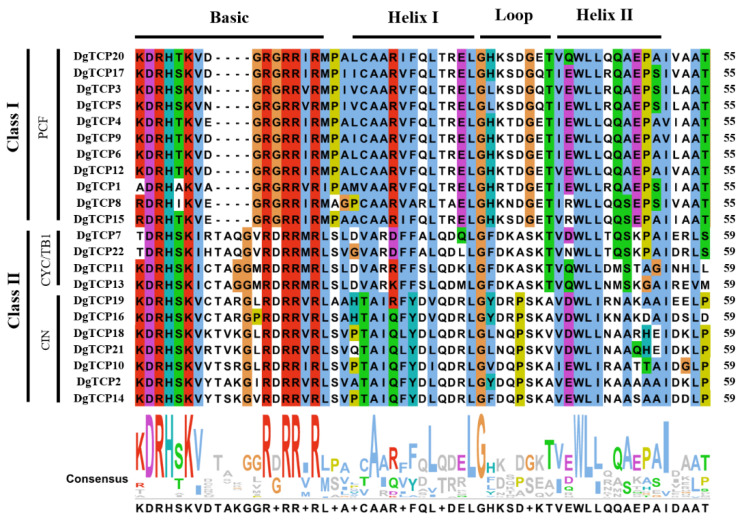
The logo and sequence alignment of TCP domains from orchardgrass. The basic helix–loop–helix structure has been marked.

**Figure 3 genes-14-00925-f003:**
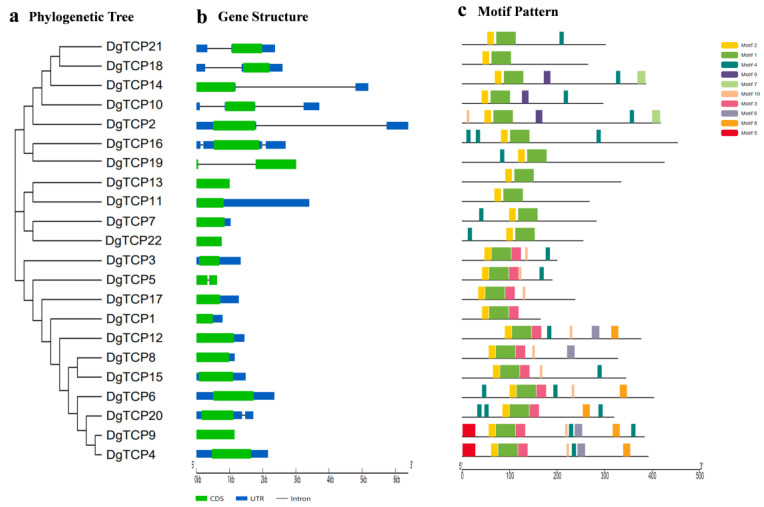
The phylogenetic tree of the orchardgrass *TCP* gene family, the gene structures, and the motifs. (**a**) The phylogenetic tree of *D glomerata* TCP proteins (*DgTCP*). (**b**) Exon–intron structures of *DgTCP* proteins. Blue squares indicate UTR, black lines indicate introns, and green squares indicate CDS. (**c**) The colored squares represent the conserved motifs of the DgTCP proteins.

**Figure 4 genes-14-00925-f004:**
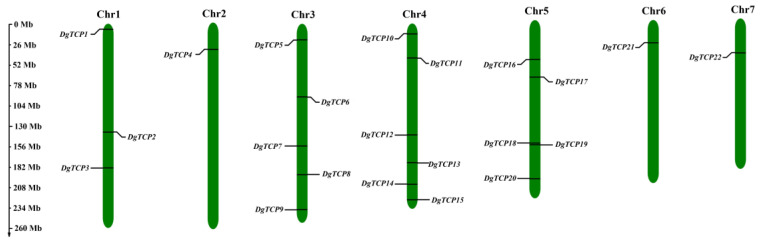
The chromosome locations of the *DgTCP* genes. The green bars represent *Dglomerata* chromosomes.

**Figure 5 genes-14-00925-f005:**
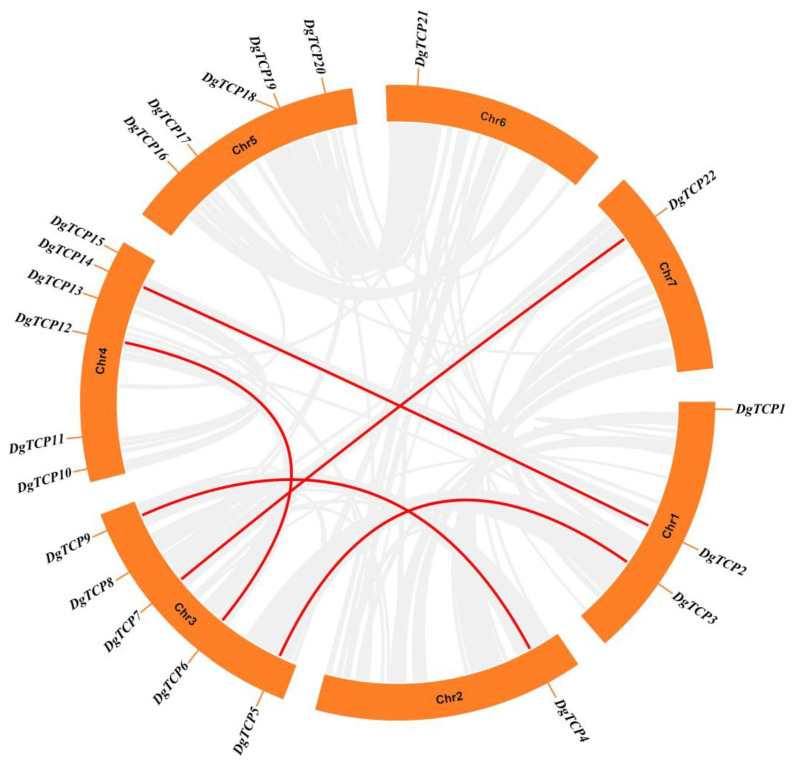
Genomic locations and segmental repeats of the *DgTCP* genes. The red lines represent five segmental duplicates of the 22 *DgTCP* genes.

**Figure 6 genes-14-00925-f006:**
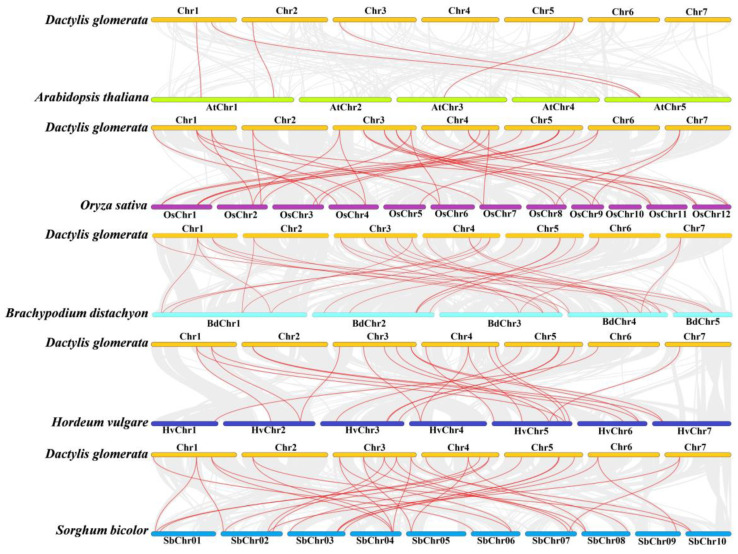
Collinearity of the *TCP* genes between *D glomerata* and *A thaliana*, *O sativa, S bicolor, H vulgare,* and *B distachyon*. The red lines in the background represent the *TCP* gene pairs.

**Figure 7 genes-14-00925-f007:**
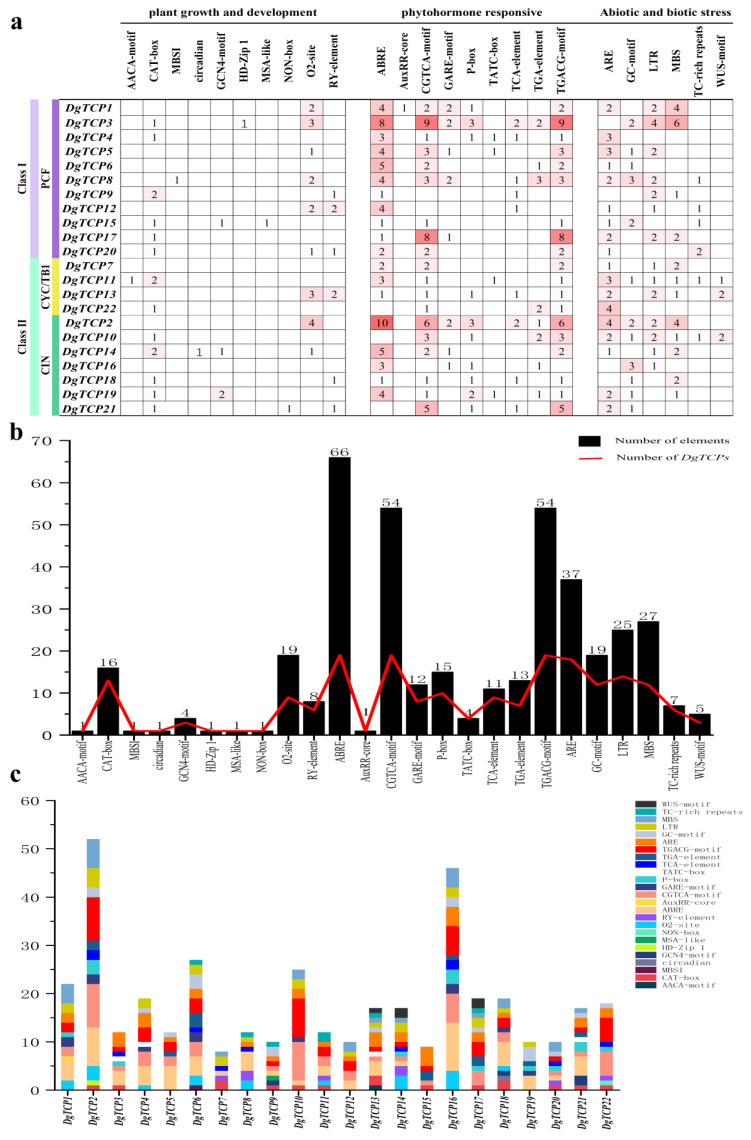
Cis-elements in the promoter regions of the *DgTCP* genes. (**a**) The cis-elements in the *DgTCP* promoter regions. (**b**) The number of *DgTCPs* and corresponding cis-elements (red line) and the total number of cis-elements in the *DgTCP* gene family (black box). (**c**) The number of cis-elements (from the promoter regions of each *DgTCP)* associated with plant growth and development, phytohormone, and stress responses.

**Figure 8 genes-14-00925-f008:**
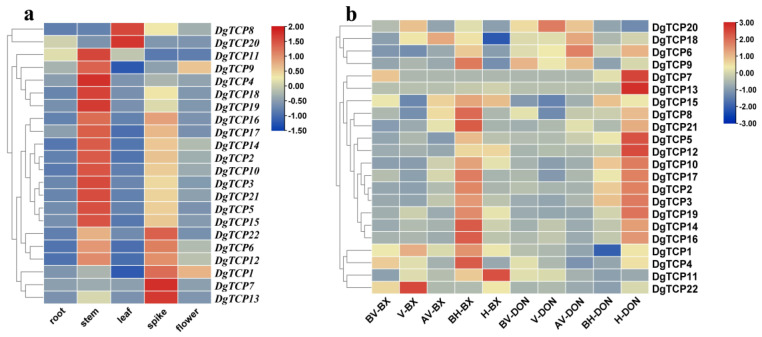
Expression profiles of 22 *DgTCP* genes in different orchardgrass tissues and developmental stages. (**a**) The 22 *DgTCPs* in different tissues. (**b**) The 22 *DgTCPs* in “Baoxing” and “Donata” at 5 developmental stages. Color changes from blue to red represent relatively low or high expression, respectively.

**Figure 9 genes-14-00925-f009:**
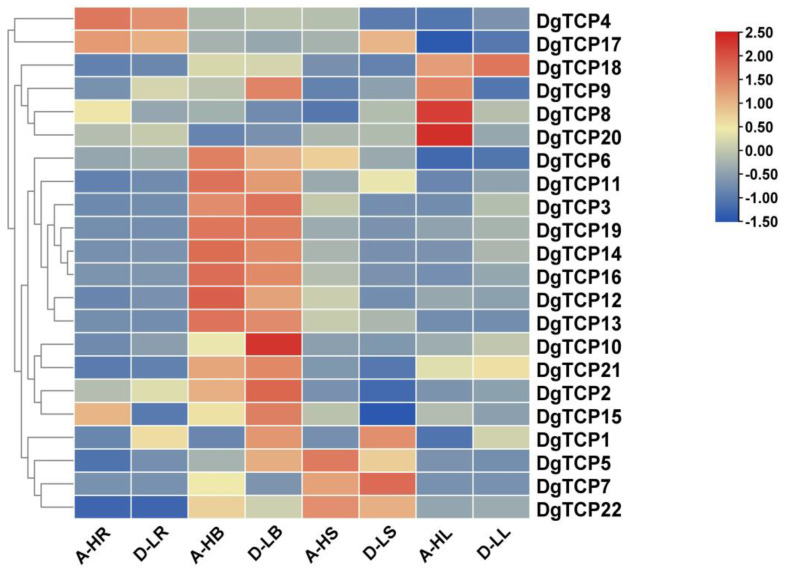
The expression of 22 *DgTCP* genes in different tissues of D20170203 (low-tillering) and AKZ-NRGR667 (high-tillering). The tissues of D20170203 include the tiller bud, D_LB; the shoot base, D_LS; the root, D_LR; and the leaf, D_LL. The tissues of AKZ-NRGR667 include the tiller bud, A_HB; the shoot base, A_HS; the root, A_HR; and the leaf, A_HL. Color changes from blue to red represent relatively low or high expression, respectively.

**Figure 10 genes-14-00925-f010:**
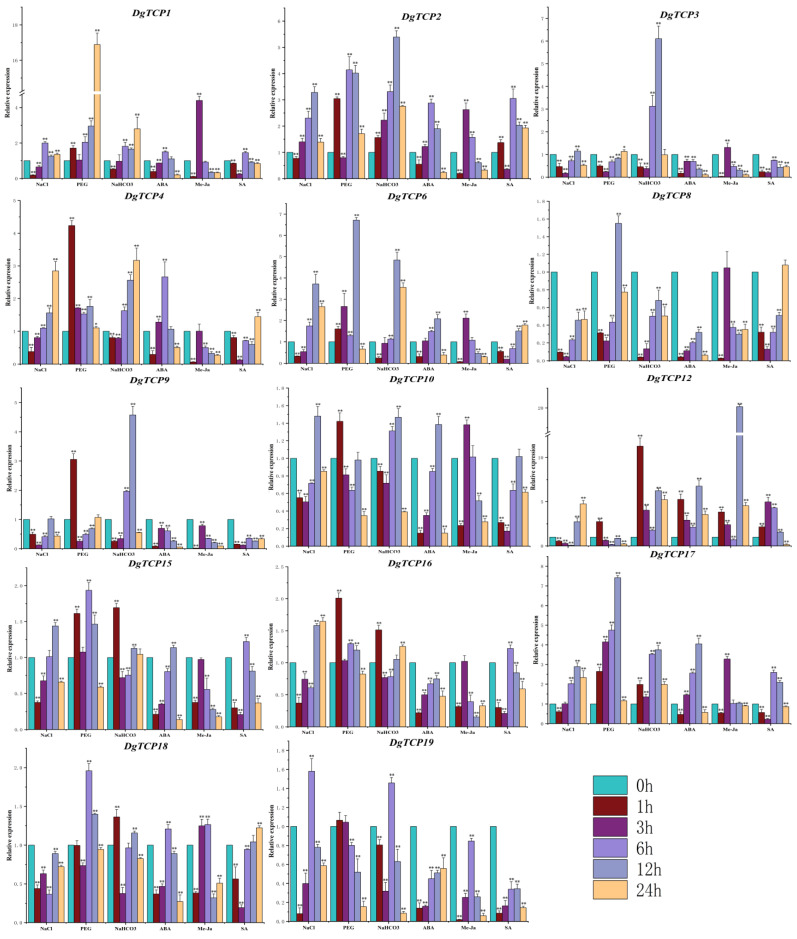
The expression of 14 *DgTCP* genes under drought (PEG), salt (NaCl), alkali (NaHCO_3_), ABA, Me-JA, and SA treatment by qRT-PCR. The error bars show the standard deviations of the three biological replicates. ****, *p* < 0.01 and ***, *p* < 0.05 show the significance of the differences between the control and treatment groups.

**Table 1 genes-14-00925-t001:** The 22 TCP genes in orchardgrass.

Gene Name	Gene ID	Chr	Protein Length (aa)	Length (bp)	Molecular Weight (kDa)	Isoelectric Point (pI)	Start	End
*DgTCP1*	*DG1C00255.1*	Chr1	165	495	17,433.51	5.18	6,629,906	6,630,695
*DgTCP2*	*DG1C03379.1*	Chr1	418	1254	44,766.77	9.34	146,031,371	146,037,761
*DgTCP3*	*DG1C04745.1*	Chr1	200	600	21,319.20	9.92	194,567,533	194,568,868
*DgTCP4*	*DG2C01041.1*	Chr2	392	1176	39,526.73	9.42	35,226,595	35,228,755
*DgTCP5*	*DG3C00669.1*	Chr3	190	570	20,377.03	9.41	20,872,684	20,873,306
*DgTCP6*	*DG3C02184.1*	Chr3	404	1212	41,896.68	5.71	98,406,794	98,409,146
*DgTCP7*	*DG3C03958.1*	Chr3	283	849	30,759.21	6.21	165,195,655	165,196,687
*DgTCP8*	*DG3C05144.1*	Chr3	328	984	33,731.69	5.09	204,201,074	204,202,229
*DgTCP9*	*DG3C06789.1*	Chr3	384	1152	40,053.43	8.49	251,633,645	251,634,796
*DgTCP10*	*DG4C00330.1*	Chr4	297	891	31,333.77	6.55	13,226,786	13,230,494
*DgTCP11*	*DG4C01109.1*	Chr4	268	804	29,570.36	9.15	45,855,850	45,859,255
*DgTCP12*	*DG4C03206.1*	Chr4	377	1131	39,248.24	7.99	150,217,992	150,219,441
*DgTCP13*	*DG4C04368.1*	Chr4	335	1005	36,236.47	9.01	187,896,786	187,897,790
*DgTCP14*	*DG4C05327.1*	Chr4	387	1161	39,895.82	9.07	217,216,581	217,221,767
*DgTCP15*	*DG4C06102.1*	Chr4	345	1035	35,534.55	5.80	238,269,030	238,270,515
*DgTCP16*	*DG5C01593.1*	Chr5	454	1362	47,382.56	6.46	55,126,146	55,128,837
*DgTCP17*	*DG5C02296.1*	Chr5	238	714	24,165.04	7.15	80,396,971	80,398,250
*DgTCP18*	*DG5C04172.1*	Chr5	265	795	28,827.16	6.90	174,582,071	174,584,669
*DgTCP19*	*DG5C04259.1*	Chr5	426	1278	45,754.54	6.09	176,990,965	176,993,974
*DgTCP20*	*DG5C05852.1*	Chr5	320	960	33,538.98	6.05	225,545,972	225,547,688
*DgTCP21*	*DG6C01132.1*	Chr6	302	906	32,382.92	6.50	31,588,652	31,591,023
*DgTCP22*	*DG7C01355.1*	Chr7	255	765	27,479.72	6.14	48,396,599	48,397,363

## Data Availability

All data generated or analyzed during this study are included in this article and its attached documents. The *Dglomerata* resources were downloaded from Dactylis glomerata genome database (http://orchardgrassgenome.sicau.edu.cn/, accessed on 20 January 2022). The genome data of the TCP transcription factor gene in *O sativa*, *S bicolor, H vulgare,* and *B distachyon* were downloaded from the Plant TFDB (http://planttfdb.cbi.pku.edu.cn/, accessed on 6 March 2022). The genome data of *A thaliana* were downloaded from TAIR (https://www.arabidopsis.org/, accessed on 12 February 2022).

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
