# Peer review of "Genome-Wide Identification, Characterization, and Expression of TCP Genes Family in Orchardgrass"

_genes, 2023, doi:10.3390/genes14040925_

Round 1

Reviewer 1 Report

Title: Genome-wide identification, characterization, and expression of TCP genes family in orchardgrass.

Abstract: The abstract is comprehensive and well explained
Introduction: The introduction contains sufficient information for the reader to understand the research problem
Materials and Methods: This is appropriate. The font type and size of the different subsections are required to made into uniform. 

Results: This section is very crisp. Author is requested to elaborate the results of subsections. This would help the reader well.

Discussion: This section is well focused.

Author Response

Dear Reviewer:

Thank you for your letter and for comments concerning our manuscript entitled “Genome-wide identification, characterization, and expression of TCP genes family in orchardgrass". Those comments are all valuable and very helpful for revising and improving our manuscript, as well as the important guiding significance to our researches. We have studied comments carefully and have made correction point-by-point. Revised portion are marked in yellow in the manuscript, and corrections to grammar are marked in blue. All revisions made to the manuscript have been marked and the responds to the reviewer’s comments are as following:

Point 1: Materials and Methods: This is appropriate. The font type and size of the different subsections are required to made into uniform.

Response : According to your comments, we have revised the font type and size of the different subsections in the Materials and Methods section of the manuscript.

Reviewer 2 Report

In this manuscript 22 DgTCP transcription factors are identified in Dactylis glomerata and their structure, phylogeny, and expression in different tissues and developmental stage are determined. The analysis of the published data was provided with a sufficient level of scientific novelty. The text of the paper is factual concrete, realistic, understandable. But, there are important flaws in the manuscript listed below:

·       In pant material and treatments section, more explanation about six abiotic stresses (drought, salt, alkali, Me-JA, ABA, and SA) is needed. Sufficient evidence should be provided for considering these six time (0, 1, 3, 6, 12 and 24h) after treatment?

·       It is not clear from the text (line 159) the three replications for qRT-PCR is biological or technical replication. If it is just technical replications, at least three biological replication is need for more accurate results.    

·       Fig. 8 and Fig. 9 need further details and explanation about the color used in the Figures.

·       In Fig. 7 and Fig. 10, the texts inside the figures are not clear, which is really incomprehensible.

Author Response

Dear Reviewer:

Thank you for your letter and for comments concerning our manuscript entitled “Genome-wide identification, characterization, and expression of TCP genes family in orchardgrass". Those comments are all valuable and very helpful for revising and improving our manuscript, as well as the important guiding significance to our researches. We have studied comments carefully and have made correction point-by-point. Revised portion are marked in yellow in the manuscript, and corrections to grammar are marked in blue. All revisions made to the manuscript have been marked and the responds to the reviewer’s comments are as following:

Point 1: In pant material and treatments section, more explanation about six abiotic stresses (drought, salt, alkali, Me-JA, ABA, and SA) is needed. Sufficient evidence should be provided for considering these six time (0, 1, 3, 6, 12 and 24h) after treatment?

Response 1: Due to the exploration of different time points and stress treatments of orchardgrass by the previous research group, a number of time points and stress treatment types with expression changes were screened out. For example, in the article by Shuai et al., the author subjected to drought, salt, alkali, and ABA stress treatments and sampled at 0h, 1h, 3h, 6h, 12h, and 24h (Shuai et al., Genome, 2020); in the article by Yang et al., the author subjected to drought, salt, and ABA stress treatments and sampled at 0h, 3h, 6h, 12h, and 24h (Yang et al., BMC Genomics, 2021). In the study of the bamboo TCP family, the author subjected to Me-JA, ABA, and SA stress treatments and sampled at 0h, 1h, 3h, 6h, 12h, and 24h (Liu et al., Frontiers in Plant Science, 2018). Therefore, we have selected these six stress treatments and six time points as the stress treatments and sampling time points for this article.

Point 2: It is not clear from the text (line 159) the three replications for qRT-PCR is biological or technical replication. If it is just technical replications, at least three biological replication is need for more accurate results.

Response 2: Thanks for your advices, we have checked the manuscript and found that the original text is unclear and has been revised. Three biological and technical replicates have been performed for all treatments in the article.

Point 3: Fig. 8 and Fig. 9 need further details and explanation about the color used in the Figures.

Response 3: According to your suggestion, we have provided detailed descriptions of the colors used in Figures 8 and 9 in lines 291-292 and 310-311 of the manuscript, respectively." Color changes from blue to red represent the relatively low or high expression, respectively".

Point 4: In Fig. 7 and Fig. 10, the texts inside the figures are not clear, which is really incomprehensible.

Response 4: We are sorry for this unclear text. And we have made adjustments to the figures 7 and 10. As shown in the following figure: 

Reviewer 3 Report

The authors stated that this study identified TCPs in orchardgrass through synthetic analysis of the D. glomerata genome (gene structure, conserved motif 82 composition, chromosomal location, and phylogenetic characteristics). Preliminary predictions of DgTCP gene evolution involved analysis of phylogenetic and gene duplication events and collinearity with other plants. Additionally, TCP expression was assessed in different tissues, developmental stages, and abiotic stresses. The results of this study will be helpful in elucidating orchardgrass adaptation to different environments, probably revealing the functions of DgTCPs. 

The results are interesting. I only suggest to enrich the introduction and discussion by addressing to the relevant papers. For example the authors can use the following paper:

Arab MM, Marrano A, Abdollahi-Arpanahi R, Leslie CA, Askari H, Neale DB, Vahdati K. (2019) Genome-wide patterns of population structure and association mapping of nut-related traits in Persian walnut populations from Iran using the Axiom J. regia 700K SNP array. Scientific Reports 9(1): 6376.

Did the authors insert this gene to the plant to study the functional genomics? If yes, please bring the results. 

1. What is the main question addressed by the research? Plant-specific TCP transcription factors regulate several plant growth and development processes. The authors report their roles in orchardgrass. 2. Do you consider the topic original or relevant in the field? Does it address a specific gap in the field? Yes, as the authors explain no know report of their roles in orchardgrass (Dactylis glomerata L) exists 3. What does it add to the subject area compared with other published material? This study determined their structure, phylogeny, and expression of TCP TFs in different tissues and developmental stages of orchardgrass. The phylogenetic tree classified the DgTCP gene family into two main subfamilies, class I and II supported by the exon-intron structure and conserved motifs. 4. What specific improvements should the authors consider regarding the methodology? What further controls should be considered? It's good if they add functional genomic results, if exist. 5. Are the conclusions consistent with the evidence and arguments presented and do they address the main question posed? Yes 6. Are the references appropriate? It is recommended to address more relevant papers. I gave some expamle. 7. Please include any additional comments on the tables and figures.

Author Response

Point 1: The results are interesting. I only suggest to enrich the introduction and discussion by addressing to the relevant papers. For example the authors can use the following paper:

Arab MM, Marrano A, Abdollahi-Arpanahi R, Leslie CA, Askari H, Neale DB, Vahdati K. (2019) Genome-wide patterns of population structure and association mapping of nut-related traits in Persian walnut populations from Iran using the Axiom J. regia 700K SNP array. Scientific Reports 9(1): 6376.

Response 1: Thanks for your advices, after reading this paper, I found that it was able to enrich the introduction and discussion of our article. We have cited this article in the introduction (line 58) and in the discussion (line 374) respectively.

Point 2: Did the authors insert this gene to the plant to study the functional genomics? If yes, please bring the results.

Response 2: Thank you very much for your insightful comments and we very much agree with this comment. At present, we mainly analyze the bioinformatics and transcriptome data to screen the genes that may be related to the regulation of growth and development and stress in orchardgrass. In the later stage, the research team will select some special TCP genes bases from the orchardgrass based on this article, and transfer them to plants for functional verification.

Reviewer 4 Report

The manuscript presents an interesting study on TCP genes family in orchardgrass.

In general, the text is well written, I have only small comments:

Line 11 “no know report of their roles in orchardgrass (Dactylis glomerata L) exists.

Line 225 and elsewhereArabidopsis” -> "Arabidopsis thaliana"

Line 239-240 two times “therefore”

Line 254 “The promoter regions of 9, 9, 15, 19, and 19”

The figures 7 and 8 have small unreadable text and low resolution which should be improved.

Author Response

Dear Reviewer:

Thank you for your letter and for comments concerning our manuscript entitled “Genome-wide identification, characterization, and expression of TCP genes family in orchardgrass". Those comments are all valuable and very helpful for revising and improving our manuscript, as well as the important guiding significance to our researches. We have studied comments carefully and have made correction point-by-point. Revised portion are marked in yellow in the manuscript, and corrections to grammar are marked in blue. All revisions made to the manuscript have been marked and the responds to the reviewer’s comments are as following:

Point 1: Line 11 “no know report of their roles in orchardgrass (Dactylis glomerata L) exists.”

Response 1: We have modified the sentence as “Nevertheless, little information is available about the TCP family in orchardgrass (Dactylis glomerata L.).”.

Point 2: Line 225 and elsewhere “Arabidopsis” -> "Arabidopsis thaliana"

Response 2: For this issue, depending on the situation, we choose to change a portion of "Arabidopsis" to "Arabidopsis thaliana"

Point 3: Line 239-240 two times “therefore”

Response 3: For lines 239-240, where "therefore" is used twice, we have changed the second "therefore" to "and" for the sake of sentence coherence.

Point 4: Line 254 “The promoter regions of 9, 9, 15, 19, and 19”

Response 4: In response to this unclear sentence, we have modified the sentence as “In several hormone-related cis-elements, the salicylic acid (TCA-element), the auxin-responsive element (AuxRR-core and TGA-element), the gibberellin-responsive element (GARE-motif, TATC-box, and P-box), the Me-JA-responsive element (CGTCA-motif and TGACG-motif) and the ABA-responsive element (ABRE) were found in the promoter region of 9, 9, 15, 19 and 19 DgTCP genes, respectively (Fig 7).”.

Point 5: The figures 7 and 8 have small unreadable text and low resolution which should be improved.

Response 5: According to your comments, we have made adjustments to the figures 7 and 8. As shown in the following figure:
